# A Systematic Review Investigating the Difference between 1 Cycle versus 2 Cycles of Adjuvant Chemotherapy in Stage I Testicular Germ Cell Cancers

**DOI:** 10.3390/medicina59050916

**Published:** 2023-05-10

**Authors:** Emanuiela Florentina Rohozneanu, Ciprian Deac, Călin Ioan Căinap

**Affiliations:** 1Department of Oncology, The Oncology Institute “Prof. Dr. Ion Chiricuţă” Cluj-Napoca, “Iuliu Hatieganu” University of Medicine and Pharmacy, 400015 Cluj-Napoca, Romania; 2Department of Oncology, “Iuliu Hatieganu” University of Medicine and Pharmacy, 400012 Cluj-Napoca, Romania

**Keywords:** stage I testicular germ cell cancers, seminomatous—STC, non-seminomatous—NSTC

## Abstract

Standard care for stage I testicular germ cell cancers (seminomatous—STC or non-seminomatous—NSTC) is orchiectomy followed by active surveillance, 1 or 2 cycles of adjuvant chemotherapy, surgery or radiotherapy. The decision on the adjuvant therapeutic approach is guided by the associated risk factors of the patient and the potential related toxicity of the treatment. Currently, there is no consensus regarding the optimal number of adjuvant chemotherapy cycles. Although in terms of overall survival, there is no proven inconsistency regarding the number of cycles of adjuvant chemotherapy, and the rate of relapse may vary.

## 1. Introduction

### Background

Testicular cancer is a rare type of tumor that accounts for about 1% of all cancers in adults [1]. In 98% of cases, the cell of origin is represented by a germ cell that failed to differentiate and continued to express its pluripotency which is afterward translated into an uncontrolled malignant growth due to the accumulation of chromosomal aberrations [2]. The uncontrolled proliferation can lead to the genesis of two histological types of testicular cancer, represented by seminoma and non-seminoma, the latter being characterized within the pathology report as being a component of embryonal carcinoma, yolk sac tumor, choriocarcinoma or immature teratoma. Orchiectomy is required for diagnosis and is also the initial therapeutic approach. Further treatment depends on the histopathological features, tumor markers reference interval (α-feto protein, β-human chorionic gonadotropin, lactate dehydrogenase) and imaging-based diagnostic analysis. Stage I testicular cancer is defined by the absence of metastasis on the retroperitoneal lymph nodes and distant organs on the CT scan. The different adjuvant therapeutic approaches depicted by one or two cycles of Carboplatin, radiotherapy of the para-aortic and ipsilateral iliac lymph nodes or active surveillance for stage I seminoma and chemotherapy using one or two cycles of BEP, RPLND or active surveillance for stage I non-seminoma provide a similar outcome in terms of overall survival [3,4]. However, it can alter the risk of relapse which can fluctuate from 15–30% for stage I seminoma to 40–50% for stage I non-seminoma [5,6] Adequate management of this stage is required, as the 5-year survival rate is close to 100% [3,4,5,6].

When chemotherapy is administered, guidelines variably recommend the administration of one or two cycles, but the specific number of cycles is not well defined yet. The aim of this systematic review is to assess the difference between one versus two administered cycles of chemotherapy in stage one testicular germ cell cancers (STC and NSTC) with reference to overall and disease-free survival, short-term and long-term toxicities. Given the high rates of survival among the patient population and the associated risk of relapse, it is necessary to identify the right approach for this stage of the disease.

## 2. Methods 

### 2.1. Literature Search Strategy 

The articles reviewed for this paper needed a systematic search conducted in PubMed (1970–December 2022) using the following key words: ’testicular neoplasms’, ’testicular cancer’, ’germinal testicular cancer’ and ’non-seminomatous tumors’ combined with ’chemotherapy’ or ’treatment’. 

### 2.2. Selection Criteria

Papers that were reviewed for this article included the following criteria:-Articles written and published in English-Evidence-based clinical practice guidelines-Randomized and non-randomized studies referring to stage I seminoma or non-seminoma treated with adjuvant chemotherapy that reported a rate of recurrence and/or survival and/or toxicity of the treatment-If any article included other stages of seminoma or non-seminoma, outcome was reported separately for stage I disease-If any article included other options of treatment for stage I seminoma or non-seminoma (surveillence, radiotherapy and surgery), outcome was reported separately for the patients who received chemotherapy

## 3. Results

### 3.1. Literature Search Results

We identified 30 articles that met the selection criteria of this paper and included 3 clinical practice guidelines, 2 randomized studies and 25 non-randomized studies. The results of the literature search are summarized in Table 1. The PRISMA guidelines were followed for drafting of this paper.

### 3.2. Clinical Practice Guidelines

Three clinical practice guidelines were reviewed for this paper and the recommendations are summarized in Table 2.

#### 3.2.1. Clinical Stage I Seminoma

ESMO [7] and EAU [8] guidelines recognize tumor size > 4 cm and invasion of rete testis as risk factors for recurrence and recommend the administration of chemotherapy in the presence of any of these factors. Moreover, the ESMO [7] guideline highlights the potential benefit of the administration of two courses of Carboplatin; however, due to insufficient evidence, it is not recommended. In contrast, the NCCN [9] guideline does not support stratification of the patients using the aforementioned risk factors due to limited evidence that fails to prove their predictive value and, therefore, recommends the administration of one or two cycles of chemotherapy whenever active surveillance is not feasible.

#### 3.2.2. Clinical Stage I Non-Seminoma

All three guidelines recommend active surveillance whenever it is feasible or one cycle of BEP in the presence of risk factors.

NCCN^8^ does not recommend two cycles given the possible associated adverse events, but EAU^9^ recognizes it as a potential option.

### 3.3. Randomized and Non-Randomized Studies

#### 3.3.1. Clinical Stage I Seminoma

When deciding on the adjuvant treatment for stage I seminoma, several studies suggested that the administration of chemotherapy should be guided by the presence of one of the two risk factors, represented by tumor size and invasion of rete testis [37]. Only one of the twelve studies exposed in Table 3 and Table 4 investigating the efficacy of adjuvant chemotherapy for clinical stage I seminoma used these factors to stratify the patients. Chemotherapy was mostly administered as an option of treatment chosen either by the physician or the patient. 

Single-cycle adjuvant Carboplatin provided satisfactory results in multiple studies [15,16,17]. In the randomized prospective study published in 2005, Oliver et al. [10] not only proved the non-inferiority of Carboplatin to radiotherapy for stage I seminoma, but after a median follow-up of 4 years, the relapse rate in the single cycle chemotherapy arm was only 4.7%.

In 2016, Dieckmann et al. [14] published a prospective non-randomized study that included a total of 725 patients with clinical stage I seminoma. Adjuvant treatment was decided by local physicians and included active surveillance, radiotherapy or chemotherapy (1 or 2 cycles of Carboplatin). After a median follow-up of 30 months, stratification of tumor size revealed that in the group of patients who received one cycle of Carboplatin, tumor dimension > 4 cm can predict a higher risk of relapse than those whose tumor is below this limit (6.8% versus 2.3%). Moreover, the risk of relapse was statistically higher for tumor sizes >5 cm (9.3%). Although the direct comparison of the number of relapses was not significantly lower for the patients treated with two courses of Carboplatin (1.5% versus 5%; *p* = 0.2048), it appears that large tumor size can predict a low efficacy of one course of Carboplatin. Similar results were obtained by Chau et al. [16] who revealed a relapse-free survival of 97.4% after one cycle of chemotherapy with Carboplatin but with a relapse rate that was nearly double for the patients with tumor size above 4 cm (5.9% versus 3.3%). Tumor size was recognized as a prognostic factor in other studies as well [22] but limited data exist regarding the role of rete testis invasion in the risk of relapse [21]. On the other hand, Oliver et al. [12] (1994) found no significant correlation between tumor size and risk of relapse which is in accordance with the results of other studies [15]. Moreover, his trial did not reveal any additional benefit in the administration of two cycles of Carboplatin when compared to one cycle, showing instead higher rates of acute and late toxicities. In contrast, the study published by Dieckmann et al. [13] (2000) revealed no relapses after two courses of Carboplatin compared to 8.6% for those treated with one cycle (*p* = 0.088). Toxicity on the reproductive system was reported, but after 20 months the median FSH levels returned to the normal range.

Reiter et al. [18] also obtained favorable results after two courses of Carboplatin without any relapses when a group of 107 patients having stage I seminoma and a toxicity profile of WHO grade I or II underwent treatment. In the study published by Steinner et al. [19], the 5-year relapse-rate after two courses of Carboplatin was 1.85% but exhibited a more hematological toxicity than previously reported: 44.4% developed thrombocytopenia, 2.8 of which was grade 4.

Aparicio et al. [21] also obtained a favorable relapse rate of 3.3% at a median follow up of 36 months and an overall 5-year survival of 100% after two cycles of Carboplatin AUC 7. Patients were stratified by tumor size (>4 cm) and invasion of rete testis, and there was a significant correlation between invasion of rete testis and relapse (DFS of 99.2% versus 91.6%; *p* = 0.0108).

#### 3.3.2. Clinical Stage I Non-Seminoma

Table 5 and Table 6 include selected trials that investigated the role of adjuvant chemotherapy in patients diagnosed with high-risk clinical stage I non-seminoma. Although the definition of high-risk disease varied, all studies had at least one risk factor in common.

Only two studies conducted a direct comparison between the outcome of patients receiving one versus two cycles of BEP. Oliver et al. [23] (2004) observed that after a median follow-up of 33 months, relapses were seen in 6.5% (3/46) of patients treated with one cycle of BEP and only in 3.6% (1/28) of patients treated with two cycles of the same regimen. No significant toxicities were reported except for permanent ototoxicity in a music teacher that led to the inability to teach. Although he was treated with two cycles of BEP, many of the studies included in this paper reported ototoxicity even after one cycle of BEP [25,27] In the SWENOTECA large prospective study, Tandstad et al. [26] stratified the patients according to the LVI invasion and offered surveillance or one cycle of BEP to those without LVI and two cycles of BEP to those LVI +. Results showed that one cycle of BEP reduces the risk of relapse by 90% to both LVI + and LVI—compared to surveillance. Furthermore, after 2 cycles of BEP relapse-free survival after a median follow-up of 60 months was 100%, with no significant additional adverse events compared to one cycle of BEP except for obstipation.

The most encouraging outcome regarding the administration of one cycle of BEP was obtained by Gilbert et al. [24]: no relapses were observed after a median follow up of 10.2 years. Similar results were published by Vidal et al. [27] and Westerman et al. [25] that obtained a relapse rate of only 2.5% and 2.7%, respectively. Results may be influenced by the small number of patients included in the studies mentioned above. Albers et al. [11] conducted a randomized phase III trial that included 191 patients comparing retroperitoneal lymph node dissection to one cycle of BEP. The authors achieved a statistically significant recurrence-free survival in favor of chemotherapy with only two relapses in the chemotherapy arm and fifteen relapses in the surgery arm after a median follow-up of 4.7 years (*p* = 0.0011). The largest and most recent prospective trial investigating the efficacy of one cycle of BEP was conducted by Cullen et al. [28] that included 246 of stage I NSGCTT. With four relapses at a median follow-up of 49 months, results showed a two year metastatic recurrence of 1.3% which is similar to the results reported for two cycles of BEP but having the advantage of low levels of serious adverse events.

Currently, there are more studies published in the literature that investigated the efficacy of two cycles of BEP than of one cycle of BEP as summarized in Table 6. In a large prospective study conducted by Cullen et al. [31], at a median follow-up of 4 years, 2 out of 114 patients had a relapse with no long-term toxicity on fertility and lung function being observed. However, the authors reported the death of a 45-year-old patient caused by a cerebrovascular event eight days after the administration of the first cycle of BEP without having hematological changes that could explain the affection. The link to the treatment was unclear. Tha data published by Maroto et al. [34] also relied on a large number of patients suffering of high-risk stage I NSGCTT (*n* = 231). After the administration of two cycles of BEP, only two reoccurrences have been observed. Regarding the toxicity on the reproductive system, out of the 19 patients who have fathered a child, only one needed to use cryopreserved semen. A total of 1.3% developed a tumor affecting the contralateral testicle. 

In a non-randomized prospective trial, Studer et al. [29] obtained a relapse-free survival of 97.5% after the administration of two cycles of BEP at a median follow up of 42 months, with only one relapse that was mature teratoma treated surgically and without late toxicities reported. The results are consistent with the data published by Pont et al. [30] that registered 2/42 relapses after two cycles of BEP for high-risk stage I NSGCTT with no significant acute or late adverse events compared to the control group. Similar rates of relapses were reported by Guney et al. [35] (4/71).

Long-term results after the administration of two cycles of BEP were published by Bohlen et al. [32], revealing only one relapse in 59 patients followed for a median time of 93 months. One case of transient nephrotoxicity, one of neurotoxicity and one of cardiotoxicity were reported. The long-term efficacy of two cycles of BEP was also studied by Chevreau et al. [33]. At a median follow-up of 113.2 months, no relapses were observed in the 40 patients receiving two cycles of chemotherapy. Two patients developed a second cancer in the contralateral testis and no impact on fertility was observed as previously reported by other studies [30,31]. Another prospective study conducted by Bamias et al. [36] reported one relapse after a median follow-up of 79 months of 142 patients.

## 4. Discussion

Historically, adjuvant chemotherapy with Carboplatin in stage I testicular seminoma was used as an alternative to radiotherapy because of the growing concerns of the side effects to this treatment [13]. Currently, adjuvant Carboplatin may be administered as an option to all patients or in the presence of risk factors (tumor size > 4 cm, invasion of rete testis). However, these risk factors have been the subject of a long debate in the past years since no prospective study has been conducted in order to validate them. In this setting, a tumor size > 4 cm was correlated with an increase in the risk of relapse in contrast to rete testis invasion that lacked evidence in most of the trials [14,16,22]. The first studies that assessed the efficacy of Carboplatin in eradicating micro-metastasis used two cycles of adjuvant chemotherapy, but subsequent evidence revealed that one cycle might be equivalent. [12]. However, one course of adjuvant Carboplatin seems to be insufficient to lower this risk when tumor size is above 4 cm [14]. 

When comparing studies that investigated the role of adjuvant BEP in stage I non-seminoma, one should be aware that different studies used different groups of risk factors, and this could be a risk of bias. One risk factor that was common in all the studies was lymphovascular invasion. The right definition of high-risk disease for stage I non-seminoma has been investigated by many trials. One of the most significant was a meta-analysis published by Vergouwe et al. [38] that examined 23 studies and analyzed a total of 2587 patients with stage I NSTC. 29% of the patients had occult metastasis diagnosed either during follow up or at retroperitoneal lymph node dissection. Several predictors for occult metastasis were identified, but the strongest was vascular invasion defined as venous and lymphatic invasion (OR, 5.2; 95% CI, 4.0 to 6.8). The presence of embryonal carcinoma, a high pathologic stage or size of the primary tumor were also statistically significantly associated with the presence of occult metastasis but with a weaker effect.

Adjuvant chemotherapy with BEP for stage I NSTC was first explored around 1990 with the argument that using more limited chemotherapy in patients with high-risk disease will restrict exposure to a higher amount of chemotherapy in case of a relapse [39]. Although the first studies published used two cycles of chemotherapy, more recent trials proved similar results with one cycle of BEP regarding relapse-free survival. Currently, there is no agreement between the use of one or two cycles of BEP. At first glance at the number of relapses reported in Table 5 and Table 6, two cycles of BEP seem to provide better results, although the difference is not obvious. The only two studies that made a direct comparison with one cycle of BEP obtained a lower number of relapses with two cycles but with no survival benefit reported [23,26]. Meanwhile, the debate is centered on the issues of dose-related toxicities from BEP. Table 7 summarizes adverse events from BEP, grouped by dose and non-dose related [39]. The acute toxicities reported in the trials mentioned in this paper were mainly hematological and gastrointestinal (nausea, vomiting). Bleomycine-induced lung injury is a well-known dose-limiting toxicity. Lung function was analyzed before and after chemotherapy describing a discreet decrease in respiratory parameters but with no symptomatic respiratory dysfunction ^32^ or pneumonitis reported in any of the studies selected for this paper. Perhaps one of the most concerning dose-related toxicity is infertility. Most of the studies reviewed for this article reported outcomes on fertility with the majority of the patients being able to conceive one or two years after the treatment. This was applicable not only to the patients that performed one cycle of BEP but also for those who performed two cycles of BEP with minimal toxicity on fertility [34]. This is in accordance with the results obtained by Bujan et al. [40] regarding the impact of chemotherapy and radiotherapy on spermatogenesis. The authors concluded that after two or fewer cycles of BEP sperm count returns to pretreatment levels after twelve months, but not after radiotherapy or more then two cycles of BEP. 

## 5. Conclusions

Taking into account the fact that the data we currently have, we suggest that all treatment options for clinical stage I testicular cancer provide similar survival outcomes and considering the potential dose-related toxicity associated with chemotherapy, we can conclude that at the moment there is not enough evidence to support the superiority of two cycles of chemotherapy instead of one.

## Figures and Tables

**Table 1 medicina-59-00916-t001:** Literature search results.

Study Type	Number [References]
Clinical practice guidelines	3 [7,8,9]
Randomized controlled studies	2 [10,11]
Non-randomized studies	25 [12,13,14,15,16,17,18,19,20,21,22,23,24,25,26,27,28,29,30,31,32,33,34,35,36]

**Table 2 medicina-59-00916-t002:** Guideline recommendations.

Guideline	Stage I Seminoma		Stage I Non-Seminoma	
	1 × Carboplatin AUC 7	2 × Carboplatin AUC 7	1 × BEP	2 × BEP
ESMO [7]	X	Recognizes better results, but limited data available	X	N/A
EAU [8]	X	N/A	X	X
NCCN [9]	X	X	X	Not indicated because of adverse events

BEP—Bleomycin, Cisplatin, Etoposid; ESMO—European Society for Medical Oncology; NCCN—National Comprehensive Cancer Network; EAU—European Association of Urology.

**Table 3 medicina-59-00916-t003:** Studies of single cycle adjuvant chemotherapy in seminomatous testicular cancer.

Study	Eligibility	Number of Patients	Chemotherapy Regimen	Median Follow-Up (Months)	Number of Relapses
Oliver et al. [12] (1994)	-	25	1 × Carboplatin AUC 7	29	0
Dieckmann et al. [13] (2000)	-	93	1 × Carboplatin 400 mg/mp	48	8
Oliver et al. [10] (2005)	Randomized	573	1 × Carboplatin AUC 7	48	27
Dieckmann et al. [14] (2016)	-	362	1 × Carboplatin AUC 7	30	18
Tanstad et al. [15] (2011)	-	188	1 × Carboplatin AUC 7	40	7
Chau et al. [16] (2015)	-	517	1 × Carboplatin AUC 7	47.2	21
Diminutto et al. [17] (2015)	-	115	1 × Carboplatin AUC 7	22.1	6

**Table 4 medicina-59-00916-t004:** Studies of 2 cycles of adjuvant chemotherapy in seminomatous testicular cancer.

Study	Eligibility	Number of Patients	Chemotherapy Regimen	Follow-Up (Months)	Number of Relapses
Oliver et al. [12] (1994)	-	53	2 × Carboplatin AUC 7	51	1
Dieckmann et al. [13] (2000)	-	32	2 × Carboplatin 400 mg/mp	48	0
Reiter et al. [18] (2001)	-	107	2 × Carboplatin 400 mg/mp	74	0
Steiner et al. [19] (2002)	-	108	2 × Carboplatin 400 mg/mp	59.8	2
Argirovic D [20] (2005)	-	163	2 × Carboplatin 400 mg/mp	48	3
Aparicio et al. [21] (2005)	T > 4 cm Rete testis involvement	214	2 × Carboplatin AUC 7	34	7
Koutsoukos et al. [22] (2016)	-	138	2 × Carboplatin AUC 6		5
Dieckmann et al. [14] (2016)	-	66	2 × Carboplatin AUC 7	30	1

T—tumor size.

**Table 5 medicina-59-00916-t005:** Studies of single cycle adjuvant chemotherapy in non-seminomatous testicular cancer.

Study	Eligibility	Number of Patients	Chemotherapy Regimen	Follow-Up (Months)	Number of Relapses
Oliver et al. [23] (2004)	VI LI Absence of yolk sac elements Presence of Undifferentiated areas Malignant teratoma Undifferentiated or malignant trophoblastic teratoma	46	1 × BEP	33	3
Gilbert et al. [24] (2006)	VI LI Absence of yolk sac elements Presence of undifferentiated elements	22	1 × BEP	122	0
Albers et al. [11] (2008)	Randomized	191	1 × BEP	56	2
Westermann et al. [25] (2008)	VI LI ≥50% embryonal carcinoma	40	1 × BEP	96	1
Tandstad et al. [26] (2009)	VI+ VI−	157 155	1 × BEP 1 × BEP	57 49	5 2
Vidal et al. [27] (2015)	VI >50% embryonal carcinoma	40	1 × BEP	186	1
Cullen et al. [28] (2020)	VI Combined seminoma + NSGCTT	236	1 × BEP	49	4

BEP—Bleomycin, Cisplatin, Etoposid; VI—vascular inasion; LI—lymphatic invasion; NSGCTT—nonseminomatous germ cell tumors of the testis.

**Table 6 medicina-59-00916-t006:** Studies of 2 cycles of BEP in non-seminomatous testicular cancer.

Study	Eligibility	Number of Patients	Chemotherapy Regimen	Follow-Up (Months)	Number of Relapses
Studer et al. [29] (1993)	VI pT > 1 Presence of embryonal carcinoma	41	2 × BEP	42	1
Pont et al. [30] (1996)	VI	29	2 × BEP	79	2
Cullen et al. [31] (1996)	Any 3 of the following criteria: VI LI Absence of yolk sac elements Presence of undifferentiated elements	114	2 × BEP	48	2
Bohlen et al. [32] (1999)	VI LI pT > 1 Presence of embryonal carcinoma	59	2 × BEP	93	1
Oliver et al. [23] (2004)	VI LI Absence of yolk sac elements Presence of Undifferentiated areas Malignant teratoma Undifferentiated or malignant trophoblastic teratoma	28	2 × BEP	33	1
Chevreau et al. [33] (2004)	VI Presence of embryonal carcinoma	40	2 × BEP	113.2	0
Maroto et al. [34] (2005)	VI Local invasion of adjacent structures Presence of embryonal carcinoma	231	2 × BEP	40	2
Guney et al. [35] (2009)	VI LI pT > 1 ≥80% embryonal carcinoma Preorchiectomy AFP ≥ 80 ng/dL	71	2 × BEP	26	4
Tandstad et al. [26] (2009)	VI+ VI−	70 2	2 × BEP 2 × BEP	60 34	0 0
Bamias et al. [36] (2011)	VI LI Invasion of tunica vaginalis, spermatic cord, rete testis or scrotal wall >50% embryonal carcinoma	142	2 × BEP	79	1

BEP—Bleomycin, Cisplatin, Etoposid; VI—vascular inasion; LI—lymphatic invasion; pT—pathologycal stage of tumor according to TNM staging.

**Table 7 medicina-59-00916-t007:** Toxicities of BEP chemotherapy [39].

Dose-Related	Non-Dose Related
Infertility Fatigue Pneumonitis or lung fibrosis Renal damage AnaemiaPeripheral neuropathy Ototoxicity Skin toxicity Reynaud’s phenomena Avascular necrosis hip	Neutropenia Alopecia Nausea andvomiting

BEP—Bleomycin, Cisplatin, Etoposid.

## Data Availability

The data analyzed in this systematic review were obtained from Pubmed electronic database.

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
