# Peer review of "A Systematic Review Investigating the Difference between 1 Cycle versus 2 Cycles of Adjuvant Chemotherapy in Stage I Testicular Germ Cell Cancers"

_medicina, 2023, doi:10.3390/medicina59050916_

Round 1

Reviewer 1 Report

On “A systematic review investigating the difference between 1 cycle versus 2 cycles of adjuvant chemotherapy in stage I testicular germ cell cancers”, Rohozneanu and col. reviewed the state of knowledge about the cost-benefits of administering 1 versus 2 cycles of chemotherapy in stage 1 testicular germ cell cancers. They conclude that there is not strong evidence to establish the right approach. The workflow and criteria for papers selection is properly described and all works are well described and tables summarization helps to follow this paper. This paper contribute to summarize the advances in this area of reseach. 

Minor comments:

1. In Abstract session it would be better to include the conclussion of this review.

2. On line 111, the sentence “predict a higher risk of relapse then those” should be corrected to “predict a higher risk of relapse thAn those”.

3. On line 113, the sentence “Although the direct comparision of the” should be replaced by “Although the direct comparison of the”.

4. In Results section the format of subtitles should be revised because it is not clear which are titles and subtitles of each section. 

Reviewer 2 Report

Rohozneanu et al have performed a systematic review of adjuvant chemotherapy in stage I testicular cancer.  This is a relevant topic, as guidelines variably suggest 1-2 cycles of adjuvant carboplatin (seminoma) or BEP (non-seminoma) and guidance about specific numbers of cycles is minimal (at least for seminoma).  The topic provides relevant information for clinicians informing these types of decisions and is a comprehensive summary of outcomes and toxicities.

Specific comments:

1. Guidelines (particularly NCCN) do not recommend adjuvant chemotherapy over surveillance, for example, in stage I testicular CA.  In fact, for seminoma, surveillance is "strongly preferred."  For nonseminoma, it is "preferred" for those without risk factors, but one of 3 options when risk factors are present.  Guidelines are careful to state that survival outcomes are not different between adjuvant BEP and surveillance (or RPLND) and it is an individual decision taking in all factors, but it is not a uniform recommendation.  Several instances in the paper read as those guidelines more uniformly recommend adjuvant chemotherapy.  I would suggest revision to reflect this equipoise in our guidelines. - Line 39 in Background, line 42 in Background, Line 89-90 in Clinical Stage I non-seminoma

2. Several typographical errors including line 158 (withot), line 315 ("Anglican" -> Anglian)

3. Careful with statement Line 246 that "better results" with 2 cycles.  Better how?  There is a numerical reduced % although this literally is just 1 recurrence case so doubt statistically significant.  The no survival difference is reported, so need to be careful that it resulted in lower # of relapses but whether that results in "better" results is unclear.

4. Conclusion should be revised to acknowledge that survival is equivalent regardless of adjuvant vs surveillance strategy employed.

Reviewer 3 Report

Review report

Overall Comments:

Overall, the manuscript is well written. However, there are some minor changes and suggestions that are mentioned above. Along with this, cross check the references in text and bibliography.

Abstract:

·         The abstract does not summarize the article’s major points. Apart from that, we usually introduce the reader through the background and the novelty of the study. So many results/numbers are confusing in the abstract. Instead of that, it is better to include only the conclusions of your study. 

Introduction:

·         In the introduction section it is important to briefly summarize all the recent knowledge.

·         Please remove old references and add some latest references.

·         Please write some lines regarding significance of this study. Why you conducted this study?

Conclusions:

·         Conclusions are not clearly described it’s better to rewrite.
